# Magnetically Tunable Adhesion of Magnetoactive Elastomers’ Surface Covered with Two-Level Newt-Inspired Microstructures

**DOI:** 10.3390/biomimetics7040245

**Published:** 2022-12-16

**Authors:** Shiwei Chen, Ziyuan Qian, Xiaojiao Fu, Xuan Wu

**Affiliations:** 1Department of Civil Engineering and Architecture, Chongqing University of Science and Technology, Chongqing 400030, China; 2Institute of Intelligent Machines, Hefei Institutes of Physical Science, Chinese Academy of Sciences, Hefei 230031, China; 3Robotics Laboratory, China Nanhu Academy of Electronics and Information Technology, Jiaxing 314002, China

**Keywords:** tunable adhesion, bionic-inspired adhesion, magnetoactive elastomer, two-level microstructure

## Abstract

As one of the new intelligent materials, controllable bionic adhesive materials have great application prospects in many fields, such as wearable electronic devices, wall climbing robot systems, and biomedical engineering. Inspired by the microstructure of the newt pad’s surface, this paper reports a bionic adhesive surface material with controllable adhesion on dry, wet acrylic, and iron sheet surfaces. The material is prepared by mixing the PDMS matrix with micron carbonyl iron powders (CIPs) and then pouring the mixture into a female mold prepared by Photo-curing 3D Printing for curing. As the mold interior is designed with a two-level microstructure array, the material’s surface not only coated a regular hexagonal column array with a side length of 250 μm and a height of 100 μm but also covered seven dome structures with a diameter of 70 μm on each column. In what follows, the adhesion force of the proposed materials contacted three different surfaces are tested with/without magnetic fields. The experimental results show that the MAEs covered with two-level bionic structures(2L-MAE) reported in this paper exhibit a stronger initial adhesion in the three types of surfaces compared to the normal one. Besides, we also found that the magnetic field will noticeably affect their adhesion performance. Generally, the 2L-MAE’s adhesion will increase with the external magnetic field. When the contact surface is an iron sheet, the material adhesion will be reduced by the magnetic field.

## 1. Introduction

Natural selection has enabled some animals to evolve the ability to move speedily on the inclined wall [1,2]. Research shows that the microstructure array covered by the feet of these animals is the key to their strong adhesion ability [3]. For instance, the hierarchical microstructure of gecko toes allows them to climb on the inclined wall [4]. The sucker microstructure of disk-winged bats allows it to climb on smooth surfaces [3]. The hexagonal microstructure of a tree frog’s pad allows it to climb on the wet wall [5]. Inspired by these phenomena, scholars have constructed similar microstructures on soft surfaces through photolithography [6], ultraprecision diamond cutting [7], polymer thin film dewetting [8], and other methods to reinforce the adhesion of those surfaces.

Nevertheless, when the adhesion ability is enhanced, the material usually becomes difficult to detach. Therefore, in order to solve this problem, how to prepare biomimetic adhesive materials with adjustable adhesive force to achieve the reversible switching between attach and detach states has become a hotspot in this research field [9,10,11,12,13]. In recent years, the research on controllable biomimetic adhesive materials has made great progress. The surface of such materials is usually covered with field-sensitive microstructure arrays [14]. When external stimuli are applied to the material, the morphology of the microstructure array is deformed. As a result, the contact area between the microstructure array and the substrate would be changed, thus changing the adhesion force in the macroscopic. For example, utilizing shape memory elastomers [9] and liquid crystal polymers [10] to fabricate the array of microstructures can make the adhesion force of bionic adhesive materials temperature-dependent.

In 2013, Gillies and his group [15] reported that when the serrated microstructure array is constructed on the surface of magnetoactive elastomers (MAE), the structure will be greatly deformed under the effect of an external magnetic field, thus changing its adhesion [16]. Compared with other control methods, magneto-responsive adhesion has the advantages of fast response, strong reversibility, and safety, making it a strong engineering application prospect. Subsequently, scholars [14,17,18,19,20,21] design and fabricate the different microstructures on MAEs’ surfaces to enhance their controllable adhesion properties.

Current Strategies for fabricating microstructures on MAEs’ surface typically include laser micromachining [22,23], self-assembling [24,25,26], and replica molding [27]. Laser micromachining enables rapid microstructuring of the pattern on MAE’s surface using remote near-infrared irradiation. However, the magnetic particles inside the MAE would be exposed to the air during the ablation process. Therefore, micro patterns fabricated by laser micromachining are easy to crack and break when subjected to external forces. The self-assembling method could only form “mountain-like” structures (micron size) on MAE’s surface by applying the external magnetic field during the curing progress. Moreover, the replica molding method requires specific approaches to release the MAE from the mold. Therefore, how to design the microsurface with high controllable adhesion and fabricate those structures with sufficient structural strength is still a hot topic in the research field.

As an ancient amphibian, newt has the ability to crawl on almost vertical dry and moist surfaces. Recent research [28,29] shows that this unique ability is attributed to the hexagonal column and hemispherical microstructure at its toe. In this study, mimicking the microsurface of the newt’s pad, this paper designed a 2-level microstructure array and used Photo curing 3D printing technology to prepare a female mold with this microstructure. Subsequently, we mixed and poured carbon iron powder (CIP) and PDMS matrix into the female mold in a certain proportion. Hence, the newt-inspired 2-level microarray is constructed at the MAE’s surface. On this basis, we first built a normal adhesion test system and selected normal MAE as the control group. Finally, the adhesion properties of 2L-MAE materials in dry and wet acrylic and iron sheets under different external magnetic fields have been studied experimentally.

## 2. Experiment Section

### 2.1. Biomimetic Structure Design

In 2013, Huang and Wang [28] found that the hexagonal cells are separated by narrow networking grooves on the newt’s pads, and the size ratio of them is around 60:1. Besides, the refs [28,29] reported that the increase of the size ratio would enhance their adhesion performance, when the newt-inspired adhesion surfaces are being prepared. However, increasing the size ratio would trigger the demolding difficulty during preparation. Therefore, the bionic two-level array structure designed in this paper is given in Figure 1b. The microstructure is divided into upper and lower parts (See Figure 1d). As shown in Figure 1a, The bottom of the structure has a side length of 250 μm. The array is composed of hexagonal columns, and the spacing of each hexagonal column in the array is 70 μm. Besides, ref [28] also found that the hexagonal structures on the newt’s pads are covered with nano-size hemispherical microstructures. Hence the upper structure designed in this work is composed of 7 domes on the lower hexagonal columns, with a dome height of 30 μm and a radius of 90 μm. One dome is located at the center of the hexagonal column, and the other six domes are equiangular and distribute 170 μm (See Figure 1c) from the column’s center.

### 2.2. Biomimetic Structures Preparation

As shown in Figure 2, the material preparation steps in this study include three steps: mold preparation, MAE mixture preparation, and bionic microstructure fabrication. The specific material preparation process can be described as follows:

Manufacturing of the Female mold manufacturing: female mold with designed structures was printed by using a Photo curing 3D printer (MicroArch S140, BMF Material Technology Inc., Shenzhen, China). Subsequently, the printed mold was carefully cleaned and separated from the substrate. Then, the mold was placed in absolute ethanol for 24 h, which could clean all the resin left in the mold. Finally, the release agent (efficient mold release agent, QQ-19, Qiqiang) was uniformly sprayed at the mold’s surface to facilitate demolding.

Preparation of the MAE mixture: PDMS and Sylgard 184 (a curing agent) was purchased from Dow Corning (MI). Moreover, CIP with an average diameter distribution between 3−7 μm was obtained from SigmaAldrich (Taufkirchen, Germany). PDMS and the curing agent were used as the base, and CIPs were used as a kind of magnetized particle. Firstly, the carbonyl iron powder (CIPs, a typical soft-magnetic material) and PDMS were mixed in a 10%, 15%, and 30% volume fraction and stirred at 200 rpm for 30 min using a mechanic mixer (IKA RW20 digital, Guangzhou, China). The mixture was placed in a vacuum for 30min to get rid of air bubbles.

Fabrication of the biomimetic microstructure: in this step, the MAE mixture was poured into the female mold. Then, the mold-containing mixture was placed in a vacuum for 60 min. After that, the mixture is cured at 70 °C for 4 h. Finally, three kinds of 2L-MAEs, including 10 vol.% (initial elastic modulus 1.41 MPa), 15 vol.% (initial elastic modulus 1.52 Mpa) and 20 vol.% (initial elastic modulus 1.60 Mpa) are fabricated.

### 2.3. Contacting Adhesion Testing

As shown in Figure 3, a contact adhesion tester (CAT) is designed and built into this works to test the tunable adhesion performance of MAEs. The CAT is equipped with a linear motion device (ZeberTech T-lsm050b, Beijing, China.), a load-sensitive sensor (Futek LRF400, 2.2N maximum range, Guangzhou, China), and an 8 mm diameter cylindrical probe which is made of 45 # steel. The MAE sample and the substrate are fixed on the cylindrical probe and a glass dish with double-sided tape, respectively. Therefore, The CAT could vertically translate the MAE towards and away from a substrate mounted over the glass dish.

In addition, several permanent magnets (Magpanda NingBo China, each magnetic intensity is around 100 mT) are placed under the glass dish to generate a uniform magnetic field in case the steel probe is closed to the substrates. Moreover, the magnetic intensity on the MAE samples is tested by an HT20 model digital tesla meter (Shanghai Hengtong Company, China). Besides, a paper gasket is placed between the glass dish and permanent magnets. The height of the paper gasket and the number of permanent magnets could be changed to ensure that the magnetic intensity in each test is the same.

Due to that, the accuracy of the mold is not perfect; there is a slight difference in height between the bionic microstructures. Therefore, a 1000 mN pressure load is applied to the MAE by the cylindrical probe in the vertical direction to ensure the microstructures are in complete contact with the substrate. During the test process, the probe approaches the substrate surface and finally retracts at a speed of 25 mm/s until the full separation of the MAE and substrate.

In this work, a 20 mm × 20 mm × 0.5 mm iron sheet (Zhongzhiyuan Company, Shanghai, China, Ra 0.8 μm) and a 20 mm × 20 mm × 3 mm acrylic plate (Xinyue Company, Shanghai, China, Ra 300 μm) in the dry and wet state are selected as the substrates. Besides, a 10 vol.% normal MAE without any surface microstructure is selected as the reference group in this work. All tests were conducted at room temperature, and all tests were repeated ten times under the same conditions. The adhesion data obtained from each test will be continuously recorded by a personal computer. Figure 3c exhibits the adhesion change curve in a test, where the highest point in the curve can be regarded as the critical pull-off load P_c_ of the material. In this paper, critical pull-off load *P_c_* is used to evaluate the adhesion performance of bionic adhesive surfaces under different contact substrates.

## 3. Results and Discussions

### 3.1. Surface Morphologies of 2L-MAE

Figure 4 gives the surface microstructure morphology of a 2L-MAE sample. From Figure 4a, we can easily find that the regular hexagonal column array has covered the surface of the 2L-MAE samples. Besides, the structural array meets the design expectation, and there are obvious equidistant gaps between them. Figure 4b shows that the hexagonal column structure has clear edges and obvious edges, which illustrates that magnetic particles do not appear at the edge of the column structure.

Figure 4c gives the SEM picture of a 2L-MAE’s surface at 150× magnification. It shows that the upper structure composed of seven micro domes exists on the hexagon microstructure, one is located in the center of the column, and the other six are located near each corner of the hexagon. Such regular distribution conforms to the design concept. We can find that the upper surface of the column structure is smooth. Figure 4d gives a detailed enlarged view of the dome structures. We can find that CI particles with an average diameter distribution between 3−7 μm protrude from the top surface of demos; hence we also see the surface of the demo is rough.

### 3.2. The Magneto Responsible Adhesion of 2l-MAE

Figure 5 gives the relationship between the magnetic field and critical pull-off load when the materials contact a dry acrylic substrate. As shown in Figure 5, when the detach process is without any magnetic fields, the adhesion of 2L-MAE is much stronger than the normal one. Moreover, the pull-off load of 2L-MAE material with a 20% particle volume ratio is as high as 79.5 mN, which is 3.8 times higher than that of normal MAE. The above results fully demonstrate that the dry adhesion property of the material is greatly enhanced by the newt-like microstructure array.

Besides, the experimental results also show that the initial adhesion force increased substantially with the CIPs’ volume fraction. We believe that the reason for this phenomenon could be explained by the JKR model [30]; the pull-off load *P_c_* in the JKR model usually can be expressed as follows:(1)$$P_{c}=\frac{3}{2}\pi r\Delta \gamma$$
where r represents the equivalent contact radius and Δγ is the work of cohesion which is related to the surface energy of those contacted materials.

Recent experiments and analytic research [31,32] show that the elastic modulus of MAE *E* could be divided into two parts: the magneto-induced modulus *E_m_* and the pure modulus *E_p_*. When the MAE is without any magnetic field, the MAE could be considered as a typical two-phase composite material; hence the raising of CIPs’ volume fraction would lead to the increase of the MAE’s elastic modulus. Consequently, the lower structure would be generated a larger press-induced deformation when the MAE contacts the substrate (see Figure 6). As a result, the equivalent contact radius r will be increased, and the pull-off load *P_c_* will increase correspondingly according to Equation (1).

In addition, different from the adhesion of normal MAE, the adhesive force of 2L-MAE will increase dramatically with an external magnetic field. When the magnetic field increases from 0 mT to 300 mT, the critical pull-off load of 2L-MAE material with a 10% particle volume ratio increases from 49 mN by 165% to 81 mN. Nevertheless, the adhesion of 2L-MAE at 15 vol.% and 20 vol.% increased to 91 mN and 115 mN, respectively, by 152% and 144%. This phenomenon of magnetic field-enhanced adhesion was also found in reference [33]. We believe that this phenomenon also causes by the magneto-induced modulus *E_m_*, which could be expressed as [31]:(2)$$E_{m}=\frac{\partial v_{m}}{\partial \varepsilon }$$
where ε is the average strain of the MAE. and $v_{m}$ represents the magnetic energy intensity. It relates to the particle volume fraction and external magnetic field. Generally, the *E_m_* is positively correlated to the particle concentrations and the magnetic intensity. Therefore, The MAE with a higher volume fraction and greater magnetic field would exhibit a stronger adhesion.

Figure 7 gives the correlation between critical pull-off load and magnetic field strength of 2L-MAE material on a wet acrylic plate. It can be found that the normal MAE can almost not attach the substrates in the wetting environment. Whereas the 2L-MAE samples still exhibit a strong adhesion. It illustrates that The Newt-inspired 2-level structure can effectively discharge the liquid on the material surface during detaching. Moreover, 2L-MAE samples with different CIPs’ volume fractions exhibit a closed adhesion when not under a magnetic field. It illustrates that the elastic modulus of the material has little effect on the adhesion performance of the 2L-MAE in the wet environment.

As shown in Figure 7, the Pull-off load of 2L-MAE material with 10 vol.%, 15 vol.%, and 20 vol.% is increased to 49 mN, 53 mN, and 55 mN, respectively, under the action of 300 mT magnetic field. 2L-MAE materials also exhibit the characteristics that the adhesion force increases with the external magnetic field in the wet adhesion state. As shown in Figure 8, the capillary bridge model (CBM) [34] could explain this phenomenon. According to the CBM model, the critical pull-off load is related to the apparent contact angle (ACA). Furthermore, when the magnetic field is applied to the 2L-MAE, the magnetized particles would interact with each other and cause uniform magneto-induced deformation. Then, the surface roughness of the dome structure would change with the external magnetic fields [35]. Therefore, the ACA of the dome structure would vary with magnetic fields. As a result, the pull-off load is increased.

Figure 9 shows changes in the critical pull-off load with the magnetic field when 2L-MAE contacts the iron sheet surface. As displayed in Figure 7, 2L-MAE materials with different volume fractions have similar critical pull-off loads, about 121 mN, in the absence of a magnetic field, which is much larger than normal MAE with a value of only 62 mN. This demonstrates that on the smooth metal surface, the bionic newt microstructure array also makes the adhesion of materials tremendously competitive.

Moreover, the surface adhesion of normal MAE samples increased with the external magnetic field. On the contrary, when 2L-MAE materials are subjected to a magnetic field, their critical pull-off load will decrease sharply with the magnetic field. Among them, the adhesive force of 20 vol.% 2L-MAE material decreases the most under the action of 300 mT magnetic field, from about 120 mN to 37 mN, while the critical pull-off load of 10 vol.% and 15 vol.% 2L-MAE will decrease to 51 mN and 43 mN respectively. The mechanism of the decline in adhesion needs to be further studied. However, this behavior matches the experiment result in references [36,37]. They found that MAEs with random surfaces would decrease their friction coefficient when applied to an external magnetic field. Therefore, we have reason to believe that the detachment force between the demo structure and the iron sheet plays a primary role in the pull-off load. As shown in Figure 10, due to the magnetization and interaction of magnetic particles under the action of a magnetic field, non-uniform deformation will be generated on the demo structure’s surface and increase its surface roughness. As a result, hence the equivalent contact area will be further reduced, resulting in the reduction of the adhesive force.

## 4. Conclusions

Inspired by the research on the microstructure of Newt’s pad, we fabricated 2L-MAE material with adjustable adhesion on a dry and wet acrylic plate and thin iron plates using the template forming method. Then, we conducted relevant experiments to explore the correlation between material critical pull-off load and external magnetic field and particle volume ratio. Research findings are listed below:In the absence of a magnetic field, the 2L-MAE exhibits a stronger initial adhesion on the three surfaces than the normal one. Besides, the three 2L-MAE with different volume fraction shows similar adhesion behaviors when they have contacted the surfaces of wet acrylic plate and iron sheet. Moreover, the 2L-MAE material with 20 vol.% exhibits the highest adhesion when they are contacted with a dry acrylic substrate.The magnetic field can significantly affect the adhesion performance of 2L-MAE samples, and the adhesion would increase with the magnetic field on the dry (up over 165%) and wet (up over 229%) acrylic plate. Furthermore, the adhesion of 2L-MAEs would decrease with the external magnetic field and achieve a 69% reduction under 300 mT magnetic field when they are in contact with the surface of an iron sheet.We suggest that the Magneto-induced modulus is the main reason that the 2L-MAE could change its adhesion when it makes contact with a dry acrylic plate under a magnetic field. Moreover, the tunable adhesion properties on a wet acrylic plate and iron sheet might be caused by the magneto-induced surface deformation on the Bionic arrays.

## Figures and Tables

**Figure 1 biomimetics-07-00245-f001:**
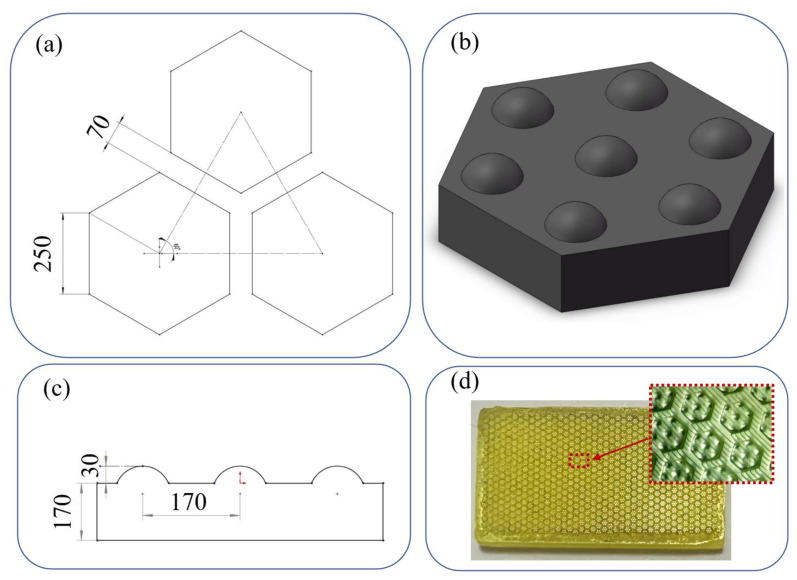
The 2-level microstructure used in the research (**a**) The vertical view of the lower structure (**b**) The 3D view of the 2-level structure (**c**)The front view of the 2-level structure (**d**) The female mold and its microstructure.

**Figure 2 biomimetics-07-00245-f002:**
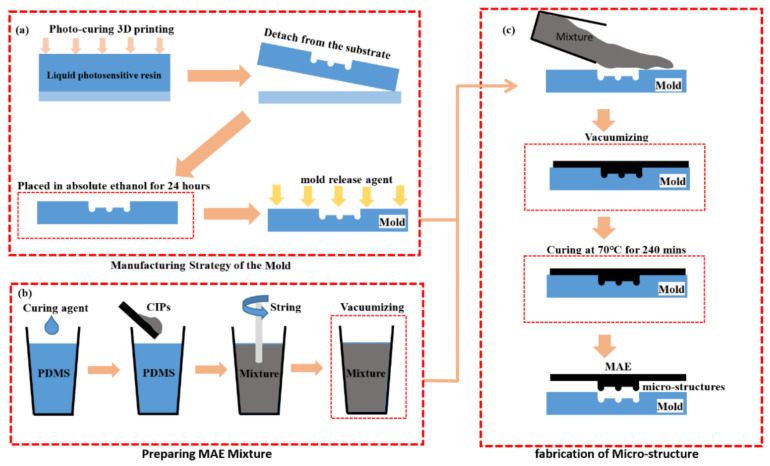
The fabrication strategy of the 2L-MAE (**a**) Manufacturing of the female mold (**b**) Preparation of the MAE mixture (**c**) Fabrication of the Biomimetic microstructures.

**Figure 3 biomimetics-07-00245-f003:**
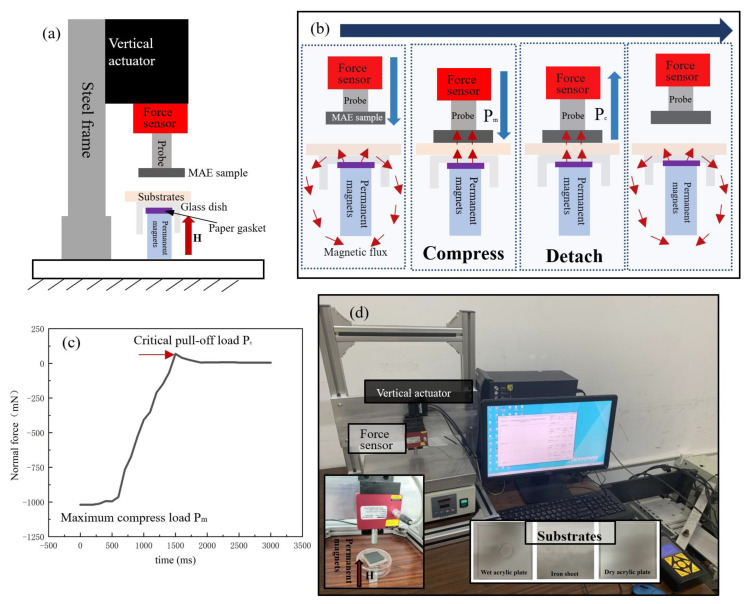
The contact adhesion tester (**a**) The schematic diagram (**b**) The testing process (**c**) The normal force vs. time curve (**d**) The picture of the contact adhesion tester.

**Figure 4 biomimetics-07-00245-f004:**
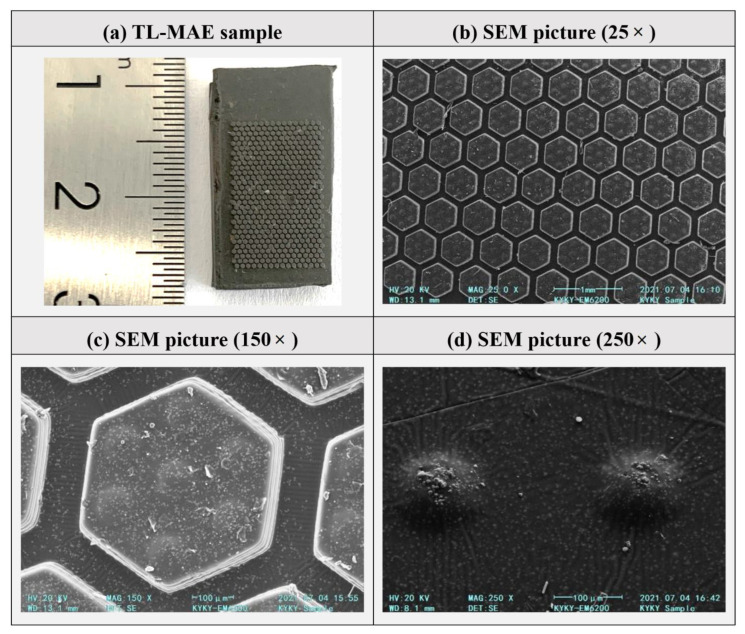
Surface morphologies of a 2L-MAE Sample. (**a**) the 2L-MAE sample (**b**) SEM picture (25×) (**c**) SEM picture (150×) (**d**) SEM picture (250×).

**Figure 5 biomimetics-07-00245-f005:**
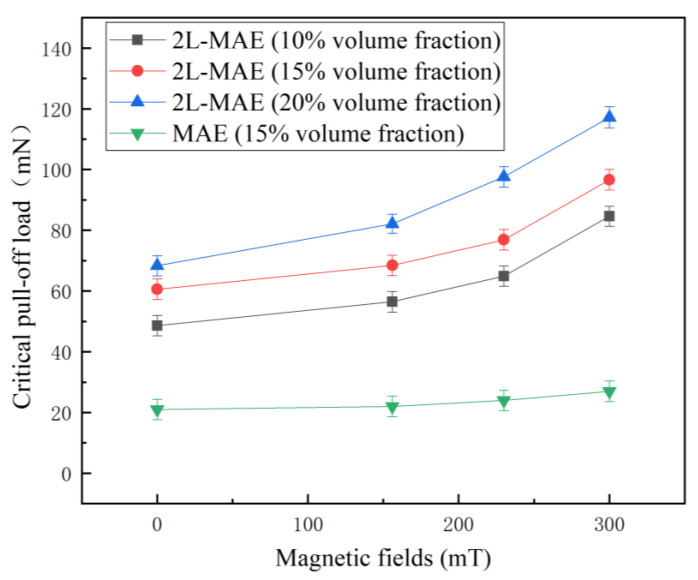
Critical pull-off load vs. external magnetic fields curves when the MAE samples detached from a dry acrylic substrate.

**Figure 6 biomimetics-07-00245-f006:**
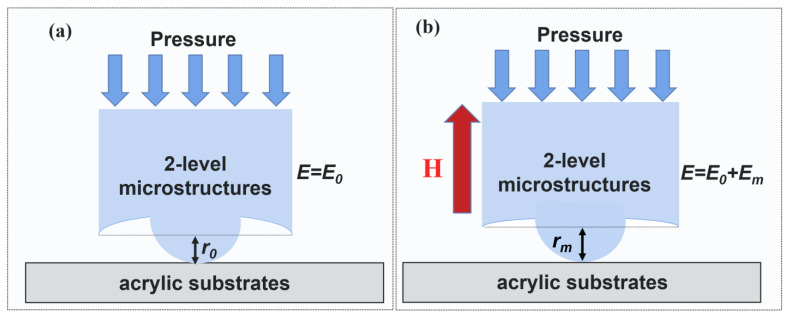
The mechanism of the magneto-responsive adhesion of 2L-MAE detached from dry acrylic substrates. (**a**) the pull-off process without a magnetic field in mesoscopic (**b**) the pull-off process with a magnetic field in microscopic.

**Figure 7 biomimetics-07-00245-f007:**
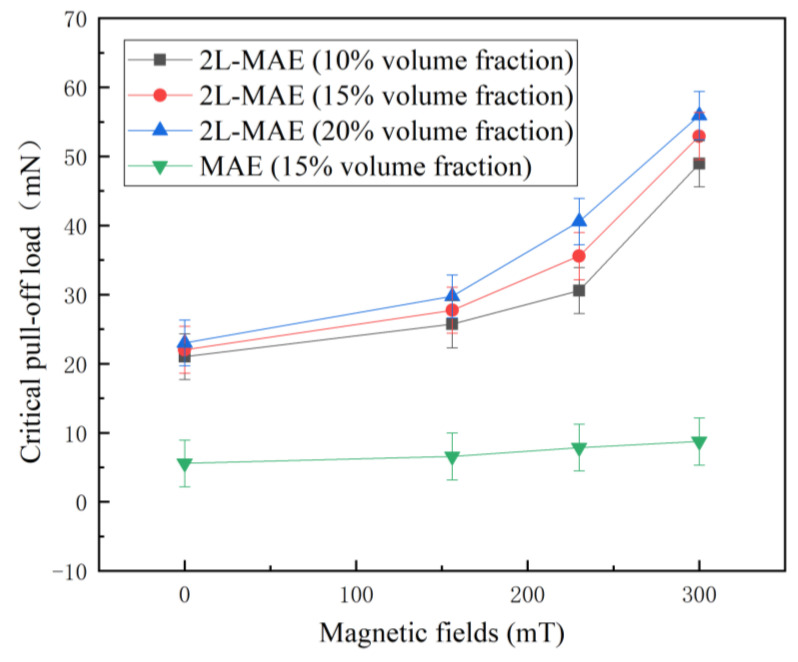
Critical pull-off load vs. external magnetic fields curves when the MAE samples detached from a wet acrylic substrate.

**Figure 8 biomimetics-07-00245-f008:**
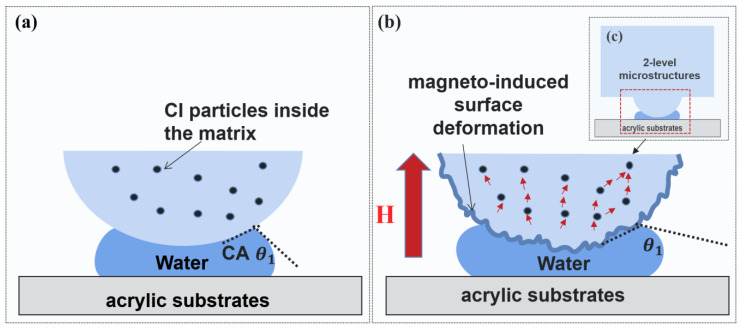
The mechanism of the magneto-responsive adhesion of 2L-MAE detached from wet acrylic substrates. (**a**) the pull-off process without magnetic field in microscopic (**b**) the pull-off process with magnetic field in microscopic (**c**) the pull-off process with magnetic field in mesoscopic.

**Figure 9 biomimetics-07-00245-f009:**
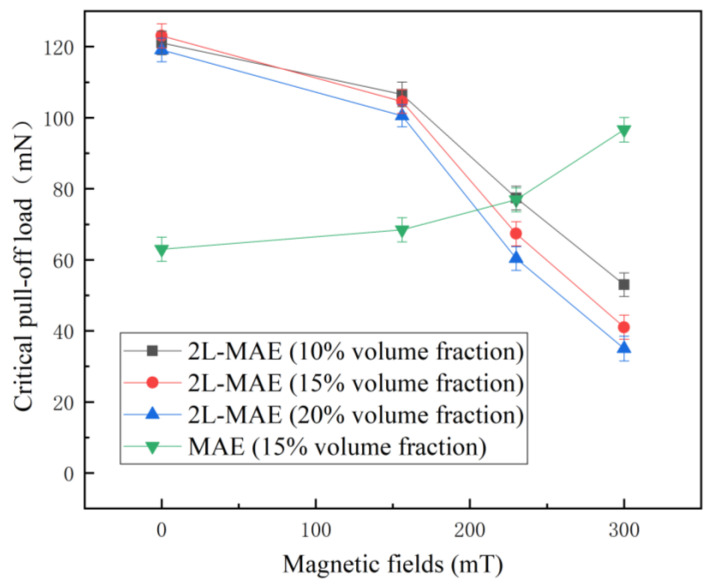
Critical pull-off load vs. external magnetic fields curves when the MAE samples detached from an iron sheet.

**Figure 10 biomimetics-07-00245-f010:**
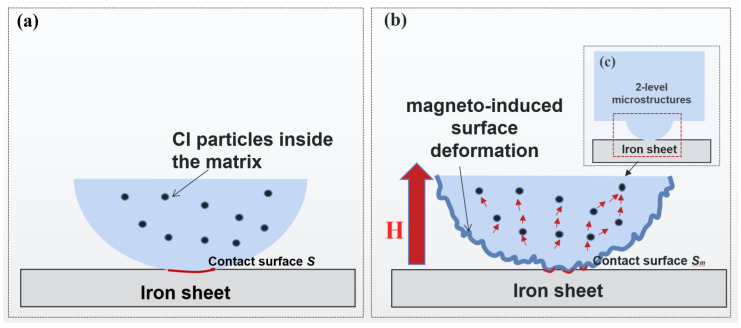
The mechanism of the magneto-responsive adhesion of 2L-MAE detached from the iron sheet. (**a**) the pull-off process without magnetic field in microscopic (**b**) the pull-off process with magnetic field in microscopic (**c**) the pull-off process with magnetic field in mesoscopic.

## Data Availability

The raw/processed data needed to reproduce these outcomes can be can be requested from corresponding author via email.

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
