# Peer review of "Magnetically Tunable Adhesion of Magnetoactive Elastomers’ Surface Covered with Two-Level Newt-Inspired Microstructures"

_biomimetics, 2022, doi:10.3390/biomimetics7040245_

Round 1
Reviewer 1 Report
The study presents the results of changing the adhesion force of a microstructured surface made of magnetoactive elastomer. To increase the effect, the surface was structured on two scales with a periodic pattern of protrusions. Authors showed that the magnitude of the magnetic field affects the amplitude of the adhesive force. I believe that the topic of the study fits nicely into the topics of the journal Biomimetics, but I suggest that before accepting it for publication, the authors significantly supplement the manuscript in the following points:
1. Considering that there is a great emphasis on the method of producing a microstructured surface, it would be necessary to present other manufacturing methods in the introduction. For example, the laser microfabrication method was recently presented (see https://onlinelibrary.wiley.com/doi/10.1002/admt.202101045 and https://www.mdpi.com/2073-4360/14/18/3883)
2. When describing the biomimetic structure, I miss an explanation of why you chose this particular geometry. Is it a reproduction of previously developed geometry, or is it the result of iterative development? A more detailed description should be added for both cases.
3. It is not clear from the description (chapter 2.2) whether you used soft magnetic particles or hard magnetic particles?
4. What does the data 100 mT mean in the sentence "During the test, several permanent magnets (magnetic field intensity 100 mT) is placed below the contact substrate to apply magnetic field to MAE samples."? The results show that the magnetic field values ​​are different from the multiple of 100 mT.
5. the results are methodologically poorly presented and explained. The authors state that the main reason is an increase in the modulus of elasticity. For this purpose, it would be good to show and discuss the effect of different concentrations of CIP particles and magnetic fields, and compare the adhesion only depending on the actual modulus of elasticity, which is higher due to the effect of the magnetic field.
6. To confirm this hypothesis, we could also test different harnesses of PDMS...
7. The ratio between the pressure force (1 N) and the adhesion force (approx. 30 mN) is very high, so I am wondering if the pressure force has any effect on the adhesion force? Is it possible that the adhesive force is constant after a certain value of the pressure force?
8. It is not clear why the trend is the opposite for steel? A better discussion is needed here.
Reviewer 2 Report
The manuscript describes fabrication of MAE surface structure with a two-level surface topography that mimics the topography of the newt pad. The fabrication process is novel and the electron microscopy-based characterization of the resulting structures is well explained. However, the measurement procedures for characterization of adhesive properties and their interpretation are very unclear and non-persuading.
1) Experimental method: In Fig 3a it is not clear where exactly is the investigated contact (between unstructured/structured MAE and acrylic/metallic surface) for which the adhesive force is measured. I assume that the investigate acrylic or iron sheet should be fixed onto the bottom of the cylindrical probe of the pressure sensor, if the adhesion between these substrates and MAE top surface is supposed to be measured. Otherwise one actually measures adhesion between MAE and probing cylinder.
On the other hand, if one would measure the force between the bottom MAE surface and the substrate laying under it, this force is expected to be predominantly the magnetic force between MAE and the permanent magnet placed under it.
One is also very sceptic on testing adhesion on iron substrate, as iron is magnetic material and so gets magnetized in the presence of magnetic field.
In all cases it is not at all clear which kind of force was really measured in the reported experiments. It is also not clear how the push-in phase takes place and why push-in process involved much higher loads than the investigate pull-off process (1000 mN versus 100 mN).
The authors should provide a more exact drawing how the sample and the substrate were oriented…
2) In Fig 4c and 4d one can see several larger objects protruding from the surface. The authors should comment how their size compares to the hemispherical surface domes.
3) The qualitative explanations of obtained results are very different and point out different aspects for different cases, i.e. every measured graph is explained by its own model. The explanation presented in Fig. 6 considers only magnetic field dependent elastic properties of MAE neglects modifications of surface roughness, while explanation shown in Fig. 8 describes all the observations with modified surface roughness. Nevertheless, in both cases one and the same MAE samples are used. While for the results shown in Fig.8 no explanation is given.
4) Minor comment: It is not clear what is “normal MAE” – is this MAE without any surface microstructure or is this MAE with 1-level (hexagonal column) structure only. I definitely recommend to compare all three materials: nonstructured MAE, MAE with 1-level structure and MAE with 2-level structure, as otherwise it is not clear which structuring level is more important.
To my opinion this manuscript is not suitable for publication, unless the authors provide a concise and clear explanation which forces exactly and how are measured in their experiments and explain more universally how the fabricated material microstructure and its properties are correlated to the obtained results.
Reviewer 3 Report
The authors report on a method to create a special relief on the surface of magnetoactive elastomers which allows to enhance the adhesion of this material to acrylic, wet acrylic and iron surfaces and to control it with the help of external magnetic fields. The results are very interesting but the explanations are not convincing and are not supported by any experimental data. In particular, elastic moduli of the obtained MAE composites and their dependence on magnetic field should be provided. Furthermore, according to Fig.3, the magnetic field seems to be inhomogeneous in the contact area between the surface and MAE film. Could the presence of field gradients explain the magnetic-field-sensitive adhesion? Finally, the text should be carefully proofread, it contains a lot of typos and poor English expressions which are hardly understandable.
Round 2
Reviewer 2 Report
The authors suitably extended the description of experimental method.
Reviewer 3 Report
The authors have answered all questions raised by the reviewer, the manuscript can be published.